# Drift and termination of spiral waves in optogenetically modified cardiac tissue at sub-threshold illumination

**Sayedeh Hussaini**[1,2,3], **Vishalini Venkatesan**[1,4], **Valentina Biasci**[5,6], **José M Romero Sepúlveda**[7], **Raul A Quiñonez Uribe**[1,3], **Leonardo Sacconi**[5,8,9], **Gil Bub**[7], **Claudia Richter**[1,3,4], **Valentin Krinski**[1,3,10], **Ulrich Parlitz**[1,2,3], **Rupamanjari Majumder**[1,3], **Stefan Luther**[1,2,3,11]*

[1]Research Group Biomedical Physics, Max Planck Institute for Dynamics and Self-Organization, Goettingen, Germany; [2]Institute for the Dynamics of Complex Systems, Goettingen University, Goettingen, Germany; [3]German Center for Cardiovascular Research, Partner Site Goettingen, Goettingen, Germany; [4]University Medical Center Goettingen, Clinic of Cardiology and Pneumology, Goettingen, Germany; [5]European Laboratory for Non-Linear Spectroscopy, Sesto Fiorentino (FI), Italy; [6]Division of Physiology, Department of Experimental and Clinical Medicine, University of Florence, Florence, Italy; [7]Department of Physiology, MGill University, Montreal, Canada; [8]Institute for Experimental Cardiovascular Medicine, University of Freiburg, Freiburg, Germany; [9]National Institute of Optics, National Research Council, Florence, Italy; [10]INPHYNI, CNRS, Sophia Antipolis, Paris, France; [11]University Medical Center Goettingen, Institute of Pharmacology and Toxicology, Goettingen, Germany

**Abstract** The development of new approaches to control cardiac arrhythmias requires a deep understanding of spiral wave dynamics. Optogenetics offers new possibilities for this. Preliminary experiments show that sub-threshold illumination affects electrical wave propagation in the mouse heart. However, a systematic exploration of these effects is technically challenging. Here, we use state-of-the-art computer models to study the dynamic control of spiral waves in a two-dimensional model of the adult mouse ventricle, using stationary and non-stationary patterns of sub-threshold illumination. Our results indicate a light-intensity-dependent increase in cellular resting membrane potentials, which together with diffusive cell-cell coupling leads to the development of spatial voltage gradients over differently illuminated areas. A spiral wave drifts along the positive gradient. These gradients can be strategically applied to ensure drift-induced termination of a spiral wave, both in optogenetics and in conventional methods of electrical defibrillation.

**\*For correspondence:**
stefan.luther@ds.mpg.de

**Competing interests:** The authors declare that no competing interests exist.

## Introduction

Emergence of reentrant electrical activity, often in the form of spiral and scroll waves, is associated with the development of life-threatening heart rhythm disorders, known as cardiac arrhythmias (*Krinski, 1968*; *Davidenko et al., 1990*; *Davidenko et al., 1992*; *Pertsov et al., 1993*). These abnormal waves stimulate the heart to rapid, repetitive and inefficient contraction, either in a periodic manner, as in the case of monomorphic ventricular tachycardia (mVT) (*Cysyk and Tung, 2008*), or in a quasi-periodic to chaotic manner, as in the case of polymorphic ventricular tachycardia (pVT) and fibrillation (*Antzelevitch and Burashnikov, 2001*). The state-of-the-art technique for controlling the dynamics of these abnormal waves involves global electrical synchronization. This is achieved by

applying high-voltage electric shocks to the heart (*Wathen et al., 2004*). However, such shocks are often associated with severe side effects, such as unwanted tissue damage (*Babbs et al., 1980*) and the development of mental disorders such as anxiety and depression in patients who experience intense pain and trauma each time the shock is delivered (*Newall et al., 2007*; *de Ornelas Maia et al., 2013*). Therefore, alternative low-energy approaches for treatment are in great demand.

One low-energy technique to control arrhythmias in the clinical setting is anti-tachycardia pacing (ATP) (*Wathen et al., 2004*). A biomedical device such as a standard implantable cardioverter defibrillator (ICD) is designed to detect the occurrence of an arrhythmia. The ATP method is based on coupling this property of the device to a local source that sends a train of electric waves in the heart to drive the spiral wave in a desired direction (*Bittihn et al., 2008*). In a finite domain, the forced drift eventually causes the phase singularity at the tip of the spiral wave to collide with an inexcitable boundary, ensuring its elimination (*Gottwald et al., 2001*). Despite its ability to control mVT and pVT (*Wathen et al., 2004*), the ATP method proves to be sub-optimal in controlling high-frequency arrhythmias and arrhythmias associated with pinned spiral waves (*Pumir et al., 2010*). Subsequent improvements by *Fenton et al., 2009*, *Luther et al., 2011*, *Li et al., 2009* and *Ambrosi et al., 2011* reduced the defibrillation threshold and fatal side effects. Further progress in the clinical implementation of these developing techniques requires a deeper understanding of the underlying spiral and scroll wave dynamics.

Recently, optogenetics has emerged as a promising tool for studying wave dynamics in cardiac tissue, overcoming some major challenges in imaging and probing (*Deisseroth, 2011*). In particular, its capabilities have been extensively used to study the mechanisms underlying the incidence, maintenance, and control of cardiac arrhythmias (*Bruegmann et al., 2010*; *Nyns et al., 2017*; *Crocini et al., 2017*; *Quiñonez Uribe et al., 2018*), and to address questions of a fundamental nature, for example the possibility to exercise control over the chirality (*Burton et al., 2015*) and core trajectories (*Majumder et al., 2018*) of spiral waves. All these studies demonstrate manipulation or abrupt termination of spiral waves by supra-threshold optical stimulation, that is stimulation that has the ability to trigger action potentials in individual cells and initiate new waves in extended media. However, very little is known about the use of optogenetics in the sub-threshold stimulation régime, which is why we have decided to investigate it in our present work. To demonstrate the drift of spiral waves using sub-threshold illumination, we use a two-dimensional (2D) continuum model of the cardiac syncytium, containing ionically-realistic representations of optogenetrically modified adult mouse ventricular cardiomyocytes at each node of the simulation domain. Sub-threshold illumination causes a shift in the resting membrane potential of optically modified heart cells without triggering action potentials. This shift affects the conduction velocity (CV) and wavelength of the propagating waves and allows spatiotemporal control of spiral wave dynamics. By applying patterned sub-threshold illumination with light intensity (LI) that is a function of space, we impose a spatial gradient on the recovery state of individual cells that make up the domain. This leads to a drift of the spiral wave along the direction of slower recovery. We show how this method can be used to ensure drift and termination of spiral waves in cardiac tissue.

## Results

In cardiac tissue, the level of electrochemical stimulation required to induce an action potential, is called the excitation threshold. Application of external stimulation below this threshold, causes small positive increase in the membrane voltage, which is insufficient to produce new waves. In this study, we use optogenetics at sub-threshold LIs to investigate the possibility of controlling spiral wave dynamics in light-sensitive cardiac tissue.

We begin with a study of the effect of uniform, global, constant sub-threshold illumination at different LI on the conduction velocity (CV) of plane propagating waves in a 2.5 cm $\times$0.25 cm pseudo domain. We find that, for electrically paced waves, CV shows a dependence on the pacing cycle length (CL) only when the CL is low (<200 ms). The CV restitution curve begins to flatten around a CL = 200 ms, for all LI (*Figure 1A*). In particular, for electrical pacing at 5 Hz, CV shows an approximate 5% decrease as LI is increased from 0 (no illumination) to 0.02 mW/mm$^2$ (*Figure 1B*). This decrease in CV may be attributed to the limited availability of $Na^+$ channels at elevated membrane voltages. In experiments, a decrease in CV was observed, with increase in LI. The reduction was two

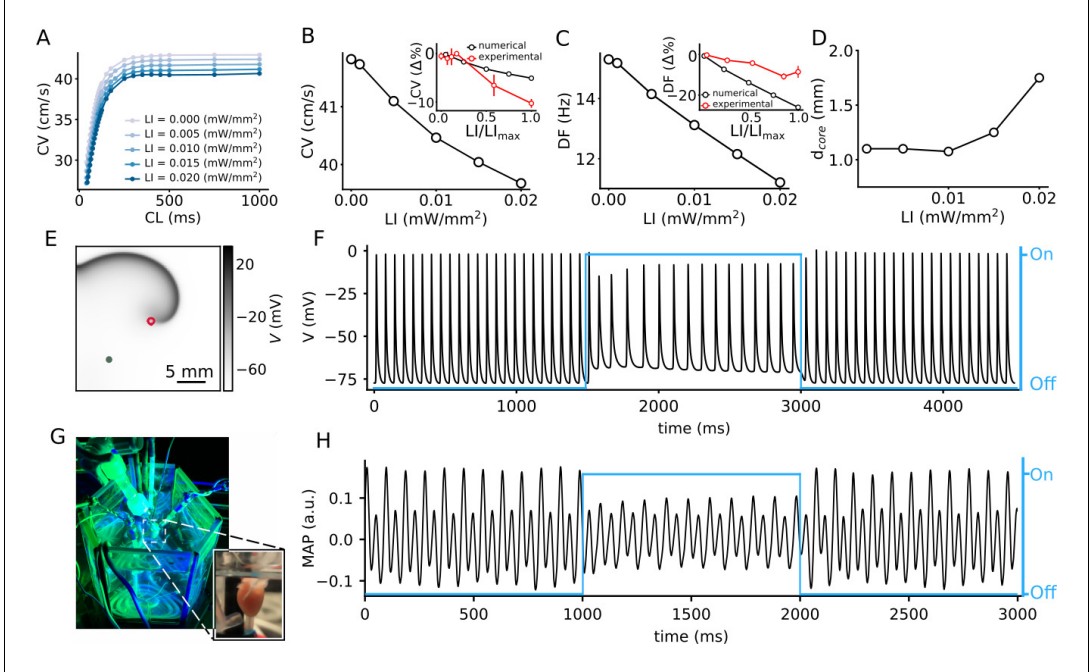

**Figure 1.** Effect of sub-threshold illumination on in silico optogenetically modified adult mouse ventricular tissue. (**A**) Conduction velocity (CV) restitution at different light intensities (LIs). (**B**) CV decreases with increase in LI, for electrical excitation waves paced at 5 Hz. Inset shows a comparison of the reduction of CV in experiments (red) and simulations (black) at different LI, relative to the unilluminated planar wave (CV reported as mean ± SEM, N = 4 with 12 trials). (**C**) Dominant frequency (DF) of a spiral wave decreases with increase in LI. Inset shows a comparison of the reduction of DF in experiments (red) and simulations (black) at different LI, relative to the unilluminated spiral (DF reported as mean ± SD, N = 2 with 14 trials). (**D**) Increase in diameter of the spiral wave core ($d_{core}$) with increase in LI. Here, core represents the circle that encloses one cycle of the spiral tip trajectory in the stationary state. (**E**) A representative snapshot of the spiral wave in a 2D simulation with a circular trajectory shown with red marker. The green marker indicates the location for extraction of the voltage timeseries in (**F**). (**G**) Our set-up of the intact mouse heart from which monophasic action potential (MAP) recordings in (**H**) were made. The blue traces in (**F**) and (**H**) illustrate the status of illumination (on/off) during the simulation or experiment.

The online version of this article includes the following figure supplement(s) for figure 1:

**Figure supplement 1.** Protocol for inducing a spiral wave in 2D.

times more than in simulations, at the highest sub-threshold LI for intact mouse hearts ($\simeq$ 0.15 mW/mm$^2$) shown as an inset of *Figure 1B*.

In 2D cardiac tissue, sub-threshold illumination seems to have a profound influence on the frequency of a spiral wave. Many theoretical and numerical studies have shown that heterogeneity in an excitable medium can cause a spiral wave to drift. This drift has a temporal component that is associated with a change in the rotation frequency of the spiral wave, and, a spatial component that is associated with the motion of the rotation center, or core, of the spiral wave (*Krinski, 1968*; *Biktasheva et al., 2010*). In the absence of light, our spiral wave rotates periodically with a temporal frequency of $\simeq$ 15.3 Hz, and a circular core trajectory. We apply uniform, global, constant sub-threshold illumination at LI $\leq$ 0.02 mW/mm$^2$, for 1500 ms. Power spectrum calculated from the voltage timeseries $V$(0.75 cm, 1.75 cm, $t$) shows periodic readouts with a single dominant frequency (DF) for each LI. We find that the DF decreases with increase in LI (*Figure 1C*). In particular, we observe a 26% reduction in the DF in simulations, in going from no illumination, to LI = 0.02 mW/mm$^2$. In experiments on the intact mouse heart, a decrease in DF reduction was observed, with increase in LI; however, the reduction was two times less than in simulations, at the highest LI for intact mouse hearts (0.015 mW/mm$^2$) shown as an inset of (*Figure 1C*). Application of sub-threshold light stimulation did not alter the general shape of the spiral tip trajectory. It remained circular at all LI considered. However, the core diameter gradually increased with increase in LI (*Figure 1D*). A representative snapshot of the spiral wave in a simulation domain with uniform, global sub-threshold

illumination at LI = 0.02 mW/mm$^2$ is shown in *Figure 1E*, with the corresponding voltage timeseries V(0.75 cm, 0.75 cm, t) in *Figure 1F*.

Similar temporal drift is observed in experiments on the intact mouse heart (*Figure 1G and H*) at LI = 0.015 mW/mm$^2$. Thus, the experimental data supports our finding that the period of the spiral can be increased in the presence of the illumination. It is important to note that the effect of the illumination is reversible, as is demonstrated by the voltage timeseries in (F) and (H), which show that the natural rotation frequency of the spiral can be restored upon removal of the light stimulus. To summarize, our results provide substantial evidence to support the change in spiral wave frequency, the so-called temporal drift, in response to uniform global constant sub-threshold illumination.

Intrinsic inhomogeneity of cardiac tissue can cause a spiral wave to drift. Such inhomogeneity can be induced using sub-threshold illumination. In order to investigate the possibility of the induction of spatial drift of a spiral wave using sub-threshold illumination, we generate a spiral wave in the non-illuminated 2D domain, and use it to define the configuration of the system at $t = 0$ s. We apply a linear gradient of sub-threshold illumination to this spiral wave. *Figure 2A* shows the spiral at $t = 2$ s, when the applied linear gradient in LI ranges from 0 mW/mm$^2$, to 0.01 mW/mm$^2$, across the length of the domain in the x-direction. In all the observed cases, the stationary spiral wave drifts toward the region with high resting membrane potential, which corresponds to the region with high LI. The inhomogeneity in the distribution of the resting membrane potential of cardiac cells across the domain establishes over time when the domain is exposed to light. We demonstrate the spatiotemporal evolution of this 'quiescent' membrane voltage (V), along the line y = 0.75 cm on the 2D domain, perpendicular to the illumination pattern (shown as a dot-dashed line in *Figure 2A*), in response to constant subthreshold illumination, in the absence of any electrical activity in the domain (see *Figure 2B*). The spatiotemporal evolution of the magnitude of the spatial derivative of the V ($dV/dx$) in shown in *Figure 2C*. Each trace represents the voltage profile along the dot-dashed line in the quiescent domain, at times corresponding to that indicated on the color bar given alongside. We have ensured that each trace in *Figure 2B* and *Figure 2C* represents the quiescent voltage profile of the domain at times which would correlate with successive turns of a spiral wave, drifting within the domain, in response to the establishing light-induced inhomogeneity. *Figure 2D–F* show the corresponding results obtained with a linear LI gradient ranging from 0 mW/mm$^2$, to 0.02 mW/mm$^2$, in the x-direction. In this case, we observe drift-induced termination of the spiral wave (see *Video 1*). It is important to note that establishment of the voltage gradients in *Figure 2B and E* impose a spatial non-uniformity in the refractory period of cells that constitute the domain. Regions with higher LI experience higher shifts in membrane voltage compared to regions with lower LI, and consequently display longer refractory period. Thus, irrespective of the range of LI used to produce the applied light gradient, the spiral wave always drifts along the direction of the longer refractory period, that is, higher LI (*Panfilov, 2009*). *Figure 2G* shows the instantaneous speed of the spiral tip (red), as a function of time, during 1000 to 1250 ms, corresponding to the spiral wave trajectory shown as inset in *Figure 2D*. The gray zones in *Figure 2G* indicate drift against the gradient, whereas, the green zones indicate the drift along the direction of the applied gradient in LI. The same figure also shows the instantaneous curvature of the tip trajectory (black). Each peak of the curvature curve corresponds to a minimum value of the speed showing drift of the spiral, against the gradient. Finally, *Figure 2H* shows the horizontal displacement of the spiral core relative to its initial position (i.e. the center of the domain), at the end of 2 s of simulation. We observe that within the given time frame spiral termination occurs only for the highest LI gradient used. The main advantage of this method is that termination can occur irrespective of the initial position of the spiral wave core.

Inspired by the success of this method to ensure drift-induced termination of spiral waves, we now work to optimize the protocol to increase the drift velocity with which the spiral wave can approach the boundary. We replace the smooth gradient illumination pattern by a step-like distribution of LI. We apply uniform sub-threshold illumination to one half of the domain such that the spiral wave core is located at the interface between illuminated, and non-illuminated regions, where the spatial derivative of the membrane voltage, in the quiescent state, is highest (see *Video 2*). *Figure 3A* shows the migration of the spiral wave from the interface toward the illuminated region during 2 s of the illumination. Typically, the drift velocity of the spiral tip in the initial phase is proportional to the slope of the gradient at the interface. However, as the spiral migrates along the direction of the illuminated region, the drift velocity decreases exponentially with time. Once the

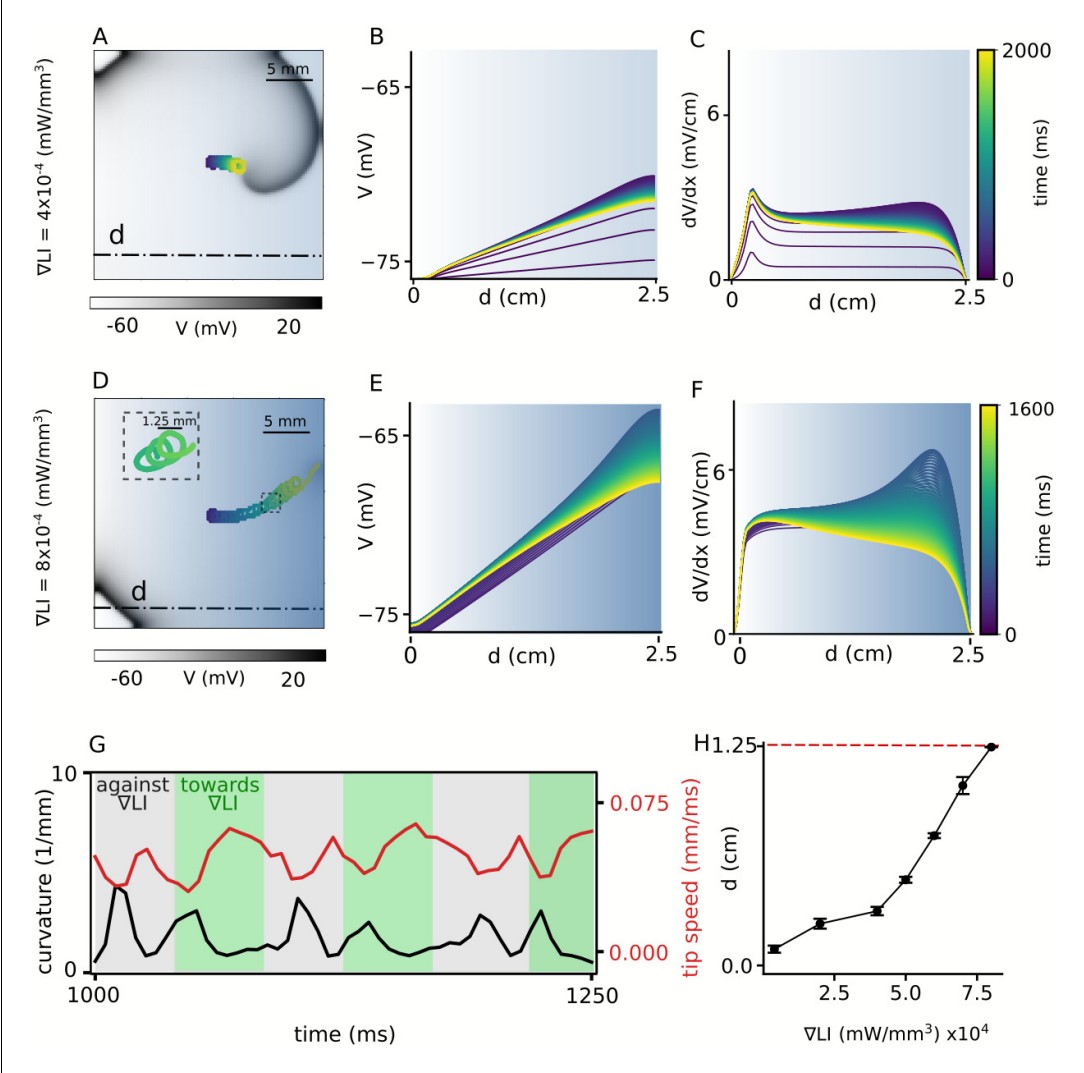

**Figure 2.** Spatial drift of a spiral wave imposed by a gradient of sub-threshold illumination. **(A)** Trajectory of a drifting spiral tip in a domain with an illumination gradient ranging from LI = 0 mW/mm² at the left boundary to LI = 0.01 mW/mm² at the right boundary. Colors indicate different times, here and B-F. **(B)** Time evolution of the voltage distribution along the dashed line indicated in **(A)**, in a quiescent domain with the same LI gradient. **(C)** Spatial derivative of the voltage distribution ($dV/dx$) along the dashed line in **(A)**, at different times, for the same applied LI gradient as in **(A–B)**. **(D)**-**(F)** show plots corresponding to **(A)**-**(C)**, but for an LI gradient ranging from 0 to 0.02 mW/mm². In this case, the spiral drifts all the way to the right boundary, within the given time frame, and terminates itself. The inset in **(D)** shows a portion of the cycloidal tip trajectory of the spiral. **(G)** Timeseries of the tip speed (red) and curvature of the spiral tip trajectory (black), as the spiral drifts along the LI gradient (green band) or against it (gray band). The profiles correspond to the part of the trajectory shown in the inset of panel D. **(H)** Increase in the maximum horizontal displacement (**d**) of the spiral core, with increase in the applied LI gradient (in mW/mm³), within the given time frame (2 s) of the simulation (displacement reported as mean ± SD, N = 10).

The online version of this article includes the following figure supplement(s) for figure 2:

**Figure supplement 1.** Spatial drift of a spiral wave imposed by a exponential pattern of sub-threshold illumination.

spiral has entered the illuminated region, it settles to a stationary state. This is due to the homogeneity of the domain away from the interface. *Figure 3B* demonstrates the spatiotemporal response of the membrane voltage to the applied light pattern in a quiescent domain (no spiral wave), along the x-axis, at y = 0.75 cm on the 2D domain, perpendicular to the illumination pattern, shown as a dashed-dot line in *Figure 3A*. It shows that the inhomogeneity of the domain at the interface causes a spiral wave to drift, but this drift is eventually inhibited as the spiral migrates away from the interface. The corresponding evolution of the magnitude of the spatial derivative of *V* is illustrated in

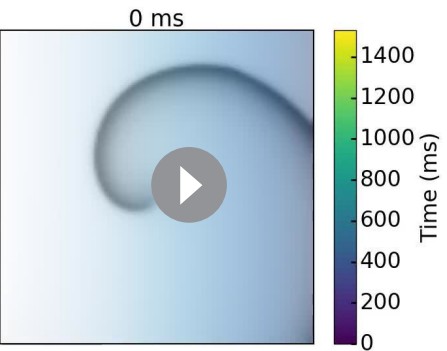

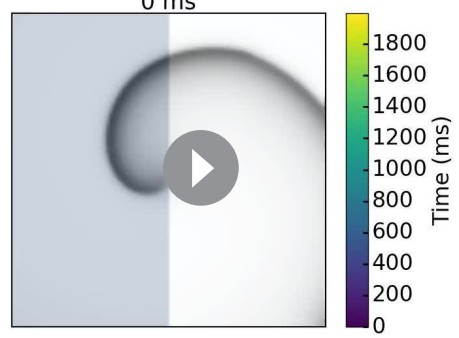

**Video 1.** Spatial drift of the spiral wave imposed by an LI gradient of $8 \times 10^4$ mW/mm$^3$. The spiral wave drifts along the illumination gradient direction. Finally the spiral wave collides to the boundary and is terminated. https://elifesciences.org/articles/59954#video1

**Video 2.** Spatial drift of the spiral wave in a domain that is partially illuminated with LI of 0.01 mW/mm$^2$. Initially, the spiral wave drifts fast toward the uniformly illuminated region, then it slows down due to the homogeneous region far from the interface of the illuminated and non-illuminated regions. https://elifesciences.org/articles/59954#video2

**Figure 3C**, with detailed analyses into the temporal growth of the peak $|dV/dx|$ and width of the $|dV/dx|$ profile at 10% maximum height (**Figure 3C**, inset). A comparison of the peak $|dV/dx|$ in **Figure 2C and F** and **3 C-D**, shows that $|dV/dx|_{max}$ in **Figure 3** is an order of magnitude larger than those in **Figure 2**. This large value of $|dV/dx|_{max}$ at the interface leads to a rapid drift of the spiral wave in the initial phase, followed by gradual deceleration (**Figure 3D**). At high LI, the tip speed shows large oscillations as the spiral moves along or against the voltage gradient imposed by the illumination. These oscillations are restricted to the width of the interface, which correlates with the width of $|dV/dx|$ at 10% peak height. However, once the spiral wave enters the illuminated region, its tip speed begins to decrease (**Figure 3D**) until it reaches a constant value. Inset of **Figure 3D** illustrates the mean squared displacement of the spiral wave tip during first 800 ms shown as shaded gray region in the speed plot. Typically, the drift velocity of the spiral tip in the initial phase is proportional to the slope of the gradient at the interface. However, as the spiral migrates in the direction of the illuminated region, the drift velocity decreases exponentially with time. Once the spiral has entered the illuminated region, it settles to a stationary state. We observe that the time required by the spiral to reach this state also depends on the intensity of the applied light. Thus, for the cases of LI > 0.015 mW/mm$^2$, the spiral wave settles to zero drift velocity (stationary state) within 2 s of observation time, whereas in others with lower LI, the drift velocity decreases to a small non-zero constant value, within the 2 s of observation time, considered. We calculated the drift velocity, for each LI and found that it decays exponentially with time as the spiral transits from the interface toward the illuminated region (**Figure 3E**). We used a function (**Equation 1**) to fit the instantaneous drift velocity as a function of time:

$$V(t) = V_0 e^{(-t/\tau)}. \tag{1}$$

We found that by increasing LI from 0.001 to 0.015 mW/mm$^2$, the initial velocity $V_0$ increases by factor of 20 and it decreases faster as manifested in the calculated values of the decay constant ($\tau$) (**Table 1**). At ≈ 0.6 cm from the interface, the influence of $dV/dx$ becomes so negligible, that the spiral establishes a stable core, bounded by a circular trajectory of its tip. A measurement of the net horizontal displacement ($d$) of the spiral tip from its initial location, at different LI during 2 s of simulation shows that $d$ increases only slightly with increase in LI (**Figure 3F**). To understand the basis for the flatenning of the $d$-LI curve, we cross-checked the mean displacements at each LI with our data on drift speed. In order to do so, we integrated the exponential fit of the instantaneous drift speed, over the time required by the spiral wave to attain stationarity, and found that the calculated displacement falls in the range of 0% to 22% tolerance of the measured numbers for $d$ in **Figure 3F**.

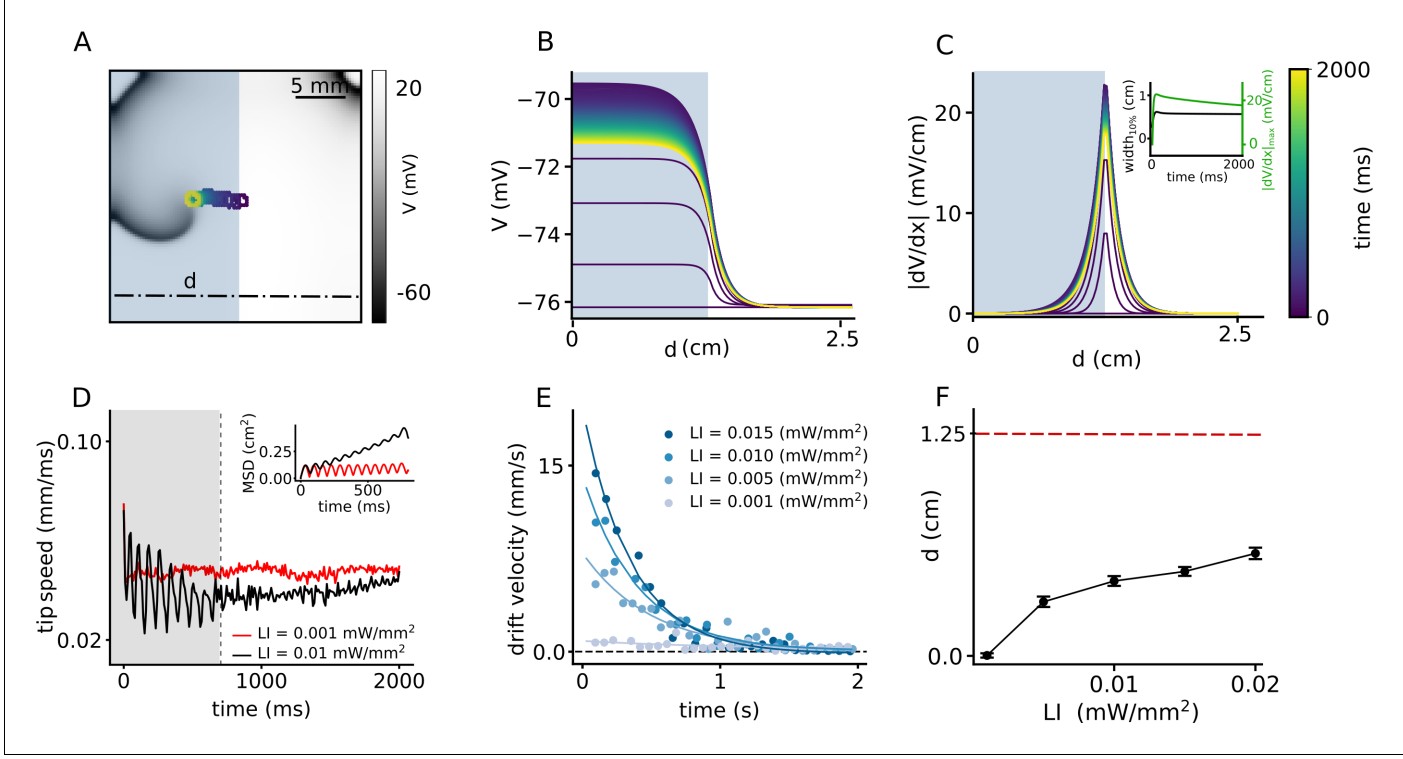

**Figure 3.** Spatial drift of a spiral wave in a domain that is partially illuminated with sub-threshold LI. (A) Trajectory of the spiral wave tip, as it drifts from the non-illuminated region to the region illuminated with LI = 0.01 mW/mm². Colors indicate different times, here and B-C. (B) Time evolution of the voltage distribution along the dashed line indicated in (A), in a quiescent domain with the same illumination. (C) Spatial derivative of $V$ ($dV/dx$) along the dashed line in (A), at different times, for the same illumination as in (A–B). Inset shows the time evolution of the distribution of $dV/dx$ at the interface between the illuminated and non-illuminated regions. The green curve shows the timeseries of the peak $dV/dx$, whereas the black curve shows the corresponding timeseries for the width of the distribution in (C). We defined 'width' as the horizontal distance between two points in the domain where $\left|\frac{dV}{dx}\right| = 0.1\left|\frac{dV}{dx}\right|_{max}$. (D) Timeseries of the spiral tip speed at LI = 0.001 mW/mm² (red) and 0.01 mW/mm² (black). Inset shows the mean square displacement profiles corresponding to the first 800 ms of illumination, shaded gray in the speed plot. (E) Timeseries of the drift speed of the spiral core at LI = 0.001 (gray), 0.005 (light blue), 0.01 (dark blue), and 0.015 mW/mm² (indigo). We observe that for any LI, drift speed decreases exponentially with time as the spiral core crosses the interface. (F) Slow increase in the maximum horizontal displacement (d) of the spiral core, with increase in LI applied to one half of the domain (displacement reported as mean ± SD, N = 10).

These values are presented in *Table 1*. A study of the trends of the exponential fits presented in *Figure 3E* shows that the higher the LI, the faster is the drift velocity across the interface. However, once the width of the interface has been crossed, drift velocity decreases rapidly to zero. Thus, the net displacement of the spiral core during the total time is comparable for different LI at high sub-threshold intensities.

To summarize, our results indicate that with a single step-like illumination gradient, even at the highest LI considered, the spiral wave does not drift sufficiently to collide with the boundary and

**Table 1.** Comparison between calculated drift-induced displacement of the spiral wave (calculated *d*), and the observed maximum displacement (*d*) at different LI, for the single-step illumination pattern.

| LI (mW/mm²) | $V_0$ (mm/s) | $\tau$ (s) | Calculated *d* (cm) | D (cm) | Tolerance (%) |
|---|---|---|---|---|---|
| 0.001 | 1 | 1.1 | 0.09 | 0.09 | 0 |
| 0.005 | 8 | 0.49 | 0.375 | 0.39 | 4 |
| 0.010 | 14 | 0.39 | 0.477 | 0.53 | 11 |
| 0.015 | 20 | 0.35 | 0.55 | 0.66 | 22 |

terminate itself. Therefore, stepwise illumination is attractive enough to draw the spiral into an illuminated area, but stabilization of the core occurs afterwards.

Our results with a single step-like gradient of illumination suggest that we need not one but a sequence of such steps to attract a spiral wave and drag it toward an inexcitable boundary to ensure its termination. Thus, we apply the following modification to the current protocol: Once the spiral wave enters an illuminated region, as in the case of half-domain illumination, we adjust the position of the interface to further pull the spiral toward an inexcitable domain boundary, resulting in continuous drift of the core. To this end, we decrease the size of the illuminated region in three steps, from half (1.25 × 2.5 cm²) to a twentieth (0.125 × 2.5 cm²) of the domain size, with a spatial interval of 0.375 cm. At each step, we apply uniform illumination with a constant pulse width. *Figure 4A* illustrates drift-induced termination of a spiral wave at LI = 0.01 mW/mm² and 500 ms pulse width (see *Video 3*). *Figure 4B* demonstrates the spatiotemporal response of the membrane voltage to the applied light pattern in a quiescent domain (no spiral wave), along the x-axis, at y = 0.75 cm on the 2D domain, perpendicular to the illumination pattern during each step, shown as a dashed-dot line in *Figure 4A*. As the illuminated area is reduced, the sharp peak in $dV/dx$ at the interface between

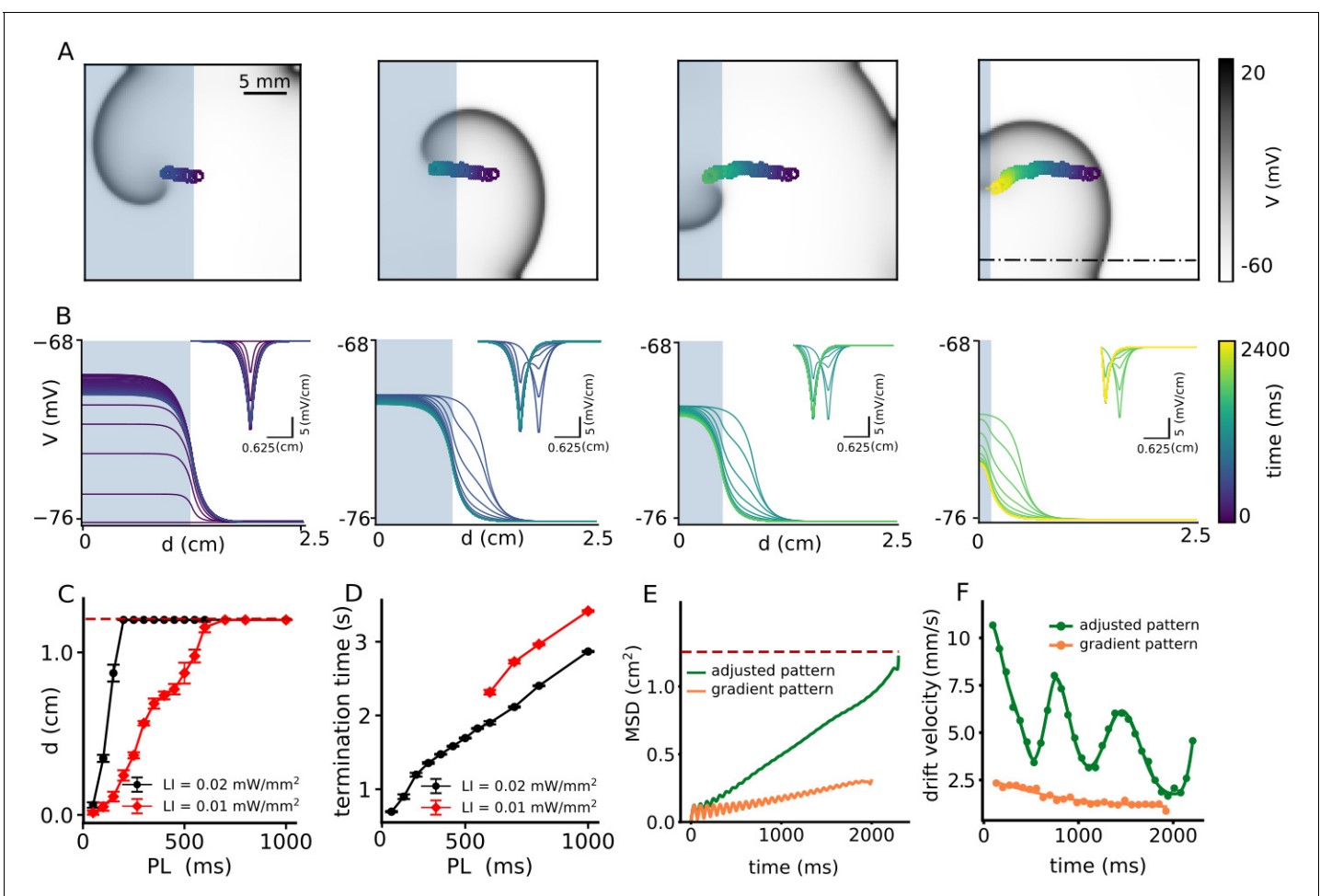

**Figure 4.** Continuous spatial drift of a spiral wave using a multi-step adjusted pattern illumination protocol. (A) Trajectory of the spiral wave tip, during different steps of the illumination protocol. (B) Time evolution of the voltage distribution and its spatial derivative $dV/dx$ (inset) for each step of the protocol, as measured along the dot-dashed line shown in the last sub-figure of panel (A). (C) Horizontal displacement $d$ of the spiral wave core at LI = 0.01 (red) and 0.02 (black) mmW/mm², respectively, at different pulse lengths (PL) (displacement reported as mean ± SD, N = 10). (D) Termination time for the cases in which the spiral drifted all the way to the boundary and annihilated itself through collision. Red and black curves represent data for LI = 0.01 mW/mm², and 0.02 (black) mW/mm², respectively (termination time reported as mean ± SD, N = 10). (E) Mean squared displacement of the spiral wave core with two different illumination protocols: multi-step adjusted pattern (green) and gradient pattern (orange). (F) Drift velocity of the spiral wave core for each case of illumination patterns, multi-step adjusted pattern (green) and gradient pattern (orange).

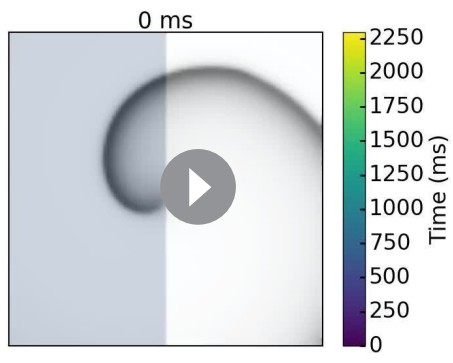

**Video 3.** Continuous spatial drift of a spiral wave using a multi-step adjusted pattern illumination with LI of 0.01 mW/mm². At each step of reducing the size of the illuminated region, the illumination was applied with a constant illumination PL of 600 ms. The spiral wave drifts continuously along with the reduction direction of the illuminated region size. Finally it collides to the boundary and is terminated.
https://elifesciences.org/articles/59954#video3

illuminated and non-illuminated regions shifts toward the boundary (*Figure 4B*). In consonance with our predictions, the spiral drifts continuously toward the boundary and terminates itself in the process. *Figure 4C* shows the maximum horizontal displacement $d$ of the spiral wave core, as it is subjected to the multi-step illumination protocol, using regional sub-threshold illumination with LI = 0.01, and 0.02 mW/mm², respectively, and a range of pulse lengths (PL) varying from 50 to 1000 ms. *Figure 4D* shows the dependence of spiral termination time on PL, for the two chosen LI, in the cases where successful termination did occur.

Finally, to compare the efficiency of spiral wave termination from different illumination protocols, such as illumination with a smooth gradient in light intensity (LI) from 0.0 to 0.01 mW/mm² (*Figure 2A*) and multi-step adjusted pattern illumination with constant LI = 0.01 mW/mm² (*Figure 4A*), we calculated the mean square displacement (MSD) of the spiral wave core during 2 s of illumination (*Figure 4E*). Our studies showed that the MSD for the case of gradient illumination (shown as an orange curve in *Figure 4E*) was ≈ 0.25 cm² at the end of 2 s, which was far from the boundary, whereas, that in the case of multi-step adjusted pattern illumination (shown as a green curve in *Figure 4E*), was 1.25 cm². The drift velocity ($V(t)$) of the spiral wave core in each case, is shown in *Figure 4F*. These results indicate that $V(t)$ is $\simeq 4x$ higher in case of illumination via multi-step adjusted pattern, than that measured with gradient illumination. Each minimum value for the case of a multi-step pattern illumination corresponds to the time at which the illumination pattern is set to the next step. Such adjustment causes the spiral wave to drift continuously toward the boundary, whereupon it terminates within 2 s of simulation. Thus, the multi-step adjusted pattern proves to be faster at effectuating drift-induced termination of spiral waves, than the gradient illumination pattern.

## Discussion

Two major factors responsible for the induction of spiral wave drift in cardiac tissue, are (i) intrinsic tissue heterogeneity (*Kharche et al., 2015*) and (ii) perturbation by an external force (*Wellner et al., 2010*; *Biktashev et al., 2011*). In the first case, the heterogeneity of the tissue may impose a gradient of refractoriness, or result in a non-stationary refractory period of the spiral wave, which would force the spiral to drift (*Krinski, 1968*; *Ermankova and Pertsov, 1988*). Heterogeneity in cardiac tissue can occur in two forms: in structure and in function. Structure-induced drift of spiral waves was studied by *Kharche et al., 2015* and *Woo et al., 2008*, among others. They found that the anatomy of the heart, along with differences in cell structure, is responsible for the induction of drift. However, spiral wave drift can also occur because of functional heterogeneities resulting from dispersion of electrophysiological parameters such as APD and CV within the tissue. This phenomenon is, in fact, more common.

Regardless of the origin of such functional heterogeneity, *Biktasheva et al., 2010* studied its effect on the dynamics of spiral waves in generic FitzHugh-Nagumo and Barkely models, with a stepwise distribution of heterogeneity, similar to what we used in our study (*Figure 3*). They showed that the center of rotation of the spiral wave can move towards one side of the step and then gradually freeze over time or continue to drift along the step with constant velocity. Our current study, which considers a realistic ionic model with 40 dynamical variables, shows similar dynamical behavior. In our case, the drift velocity is not constant. The spatial profile of the light-induced voltage gradient allows the spiral to drift with high speed while crossing the interface (see *Figure 3E*). However, as

the spiral leaves the interface, the drift velocity gradually decreases until it reaches a very small positive value at large distances from the interface within the time frame of our simulations.

Of particular interest is the drift velocity trend for different LI and $\nabla$LI. We observe that the spiral wave drifts more slowly at small values of $\nabla$LI, compared to large $\nabla$LI (*Figure 2H*). This can be explained by studying the general dynamic behaviour of the spiral wave at different LI. *Figure 1D* shows that at small LI (<0.01 mW/mm$^2$), the properties of the spiral core (e.g. $d_{core}$) are unaffected by the applied illumination. Thus, the application of a light gradient at small $\nabla$LI (<4×10$^4$ mW/mm$^3$ in *Figure 2H*), has a negligible effect on the dynamics of the spiral, resulting in very slow drift. On the contrary, at LI > 0.01 mW/mm$^2$ (*Figure 1D*), $d_{core}$ increases rapidly, leading to a strong decrease of the rotation frequency of the spiral wave. This means that the spiral now needs a little more time ($\tau$) to complete a single cycle of its rotation, before it can move one step ($L_{step}$) in space. It should be noted, however, that the rapid increase of $d_{core}$ causes a corresponding increase of $L_{step}$, which more than compensates for the increase of $\tau$. So the drift velocity effectively increases.

If, on the other hand, the case is considered with a single step of illumination (*Figure 3F*), the maximum displacement ($d$) of the spiral wave core saturates with increase of LI. This can be explained by looking at the spatiotemporal distribution of $dV/dx$ at the interface between the illuminated and non-illuminated regions. Our studies show that when light is applied to one half of the domain, a spatial profile of $|dV/dx|$ is established, which, over time, evolves such that the peak height decreases and width increases to saturation values. Both $\left|\frac{dV}{dx}\right|_{max}$ and width of $|dV/dx|$, at the instant of first illumination, increases with the LI. The gradual spatiotemporal evolution of the $|dV/dx|$ profile ensures nonlinearity in the drift velocity, which shows an exponential decrease over time. Once the spiral leaves the zone of influence of the interface, drift either stops, or becomes constant and occurs in the direction parallel to the interface. The net displacement of the spiral core can be calculated by integrating the drift velocity over time as the spiral crosses the interface. Drift velocity for high LI ($v_{drift, \sim high}$) is initially larger than that for low LI ($v_{drift, \sim low}$). However, because of the nature of the spatial profile of $|dV/dx|$, ($v_{drift, \sim high}$) decreases at a rate that is much faster than ($v_{drift, \sim low}$), such that, ($v_{drift, \sim high}$) drops to zero sooner than ($v_{drift, \sim low}$). Consequently, the corresponding displacement of the spiral core in the different cases with high LI come comparable. Hence the flatenning of the curve in *Figure 3F*.

In our simulations, we observed that the effective drift direction of a spiral wave always follows the direction of increasing light intensity. This is consistent with the findings of *Davydov et al., 1988*. Similar observations were also made by *Markus et al., 1992*, in light-sensitive Belousov-Zhabotinsky (BZ) reactions, where they used experiments to demonstrate positive phototaxis of a spiral wave, in the presence of a gradient of illumination. In our study we use this feature strategically to remove spiral waves from the domain by drift-induced collision with the boundary in favor of termination.

While previous studies by *Feola et al., 2017* and *Majumder et al., 2018* have proven that full spatio-temporal control over the dynamics of a spiral wave can be achieved by manipulating its core with supra-threshold illumination, the efficacy of their respective methods at sub-threshold illumination, remained untested. In this study, we exploited the power of regional sub-threshold illumination, to manipulate spiral wave dynamics with or without prior knowledge about the location of the spiral core.

Spiral wave drift can be investigated by many different approaches; each approach has its advantages and disadvantages and is designed based on the specific parameters of the system. The general conclusion is that controlled drift can lead to effective termination of spiral waves. It is therefore important to have efficient control over spiral wave drift. To this end, it is essential to develop a deeper understanding of the dynamics of drift. It has been established so far, that light-sensitive BZ reaction is the easiest-to-control excitable system for the study of spiral wave drift in experiments. Thus, optogenetics, which is the analogous tool for light-sensitivity in cardiac tissue, is expected to hold great potential in studies that involve exercising control over spiral wave dynamics in the heart (*Braune and Engel, 1993*). Our results prove the validity of this statement by demonstrating the use of optogenics to study and control spiral wave drift in 2D cardiac tissue.

Finally, controlled spiral wave drift finds its main application in optogenetics defibrillation. Current techniques for such defibrillation use global or structured illumination patterns applied to the epicardial surface of the heart. Due to the poor penetration of light into cardiac tissue, most of the applied

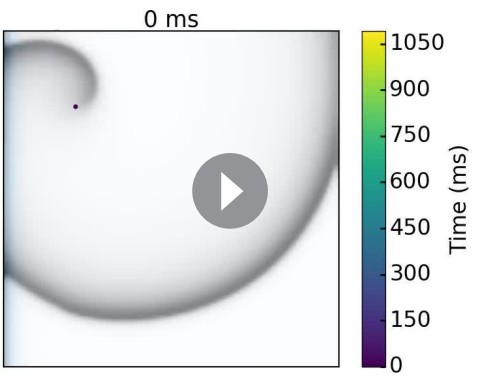

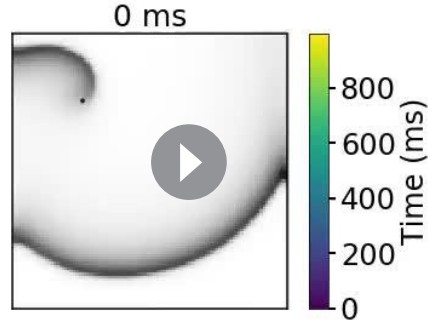

**Video 4.** Drift of a spiral toward the illuminated region with an exponential decay pattern with LI 0.07 mW/mm². https://elifesciences.org/articles/59954#video4

**Video 5.** A spiral wave rotation with no illumination pattern.
https://elifesciences.org/articles/59954#video5

light is scattered or absorbed before it can reach the endocardium. This is considered a major limitation of optogenetics, as the applied supra-threshold illumination cannot be expected to affect sub-surface electrical activity in the heart wall. However, ex vivo studies on small mammalian hearts consistently demonstrate the success of optogenetic defibrillation, without providing a clear mechanism for the same (*Bruegmann et al., 2016*). Some studies try to explain the mechanism behind this success, using the critical mass hypothesis (*Zipes et al., 1975*). According to this hypothesis, a spiral (scroll) wave requires a minimum area (volume) of excitable tissue for its sustainment. By applying supra-threshold light to the surface of the epicardium, one can effectively reduce available area (volume) of excitable tissue to below the threshold requirement for spiral sustainability, thereby forcing the wavefront of the spiral (scroll) wave to collide with its waveback, resulting in its termination. However, our studies postulate an alternative theory. We propose that the application of light to the epicardial surface effectively leads to a transmural illumination gradient within the heart wall, with both sub-threshold and supra-threshold illumination régimes. Our study shows that a linear gradient of pure sub-threshold illumination has the potential to induce a drift of a spiral wave into the region of higher illumination (i.e. the epicardial surface in a transmural section of the heart wall), thereby protecting the internal tissue from hidden electrical activity. Once such activity is drawn out to the surface by the induced drift, it can be terminated using global supra-threshold illumination, which then ensures electrical synchronization.

Thus, our study provides new mechanistic insights into the theory of successful optogenetic defibrillation in animal hearts, where application of light to the surface produces a gradient of illumination within the heart wall. This gradient is typically exponential. However, to a first approximation, we have treated it as a linear gradient to study the effect of subthreshold illumination in a simple system. *Saito et al., 2017* shows the approximate space constant for exponential decay of blue light in cardiac tissue is 0.6 mm. Studies by *Bruegmann et al., 2016* demonstrate attenuation of light at a depth of 1 mm beneath the surface of the cardiac tissue, which limits the effect of supra-threshold illumination to only a few layers below the surface. *Video 4* shows drift of the spiral wave toward the illuminated region when the wave is located at the neighboring of the illumination gradient with a exponential decay pattern of illumination in the cardiac tissue. *Figure 2—figure supplement 1A and B* show time evolution of the voltage and spatial derivative of the voltage in a quiescent domain, respectively. *Video 5* shows a spiral wave rotation in a circular trajectory where there no illumination pattern. This hypothesis can be also extended to the conventional defibrillation methods. Applying an electric field to excite the heart tissue results in the development of a transmural depolarization gradient (*Dosdall et al., 2010*). The functional heterogeneity caused by these depolarization gradients may force spiral waves to drift. Such a drift occurs in the direction of the positive gradient, resulting in the emergence of the spiral cores on the surface, where they are eliminated through synchronisation.

We propose to test this hypothesis in cell culture experiments by projecting a gradient sub-threshold light pattern on a monolayer of optogenetically modified mouse cardiac cells. Such a

pattern can be easily generated by using a diffuse light source that is slightly translated perpendicular to the field of view of the microscope. As a next step, one can try illuminating the epicardium in a transmural slice of the mouse heart using a ring of LEDs (so that the slice is illuminated uniformly, along the periphery). Using supra-threshold light on the surface of the epicardium, one can observe the dynamics of the spiral wave as in the presence of the gradient sub-surface heterogeneity. This hypothesis can be also extended to the conventional defibrillation methods. Applying an electric field to excite the heart tissue results in the development of a transmural depolarization gradient (*Dosdall et al., 2010*). The functional heterogeneity caused by these depolarization gradients may force spiral waves to drift. Such a drift occurs in the direction of the positive gradient, resulting in the emergence of the spiral cores on the surface, where they are eliminated through synchronisation.

## Materials and methods

### Numerical study

Electrical activity in single cardiac cells was modeled according to *Equation 2*. Here, $V$ is the transmembrane voltage that arises from ionic gradients that develop across the cell membranes.

$$\frac{dV}{dt} = -\frac{I_{ion} + I_{stim}}{C_m} \tag{2}$$

The total ionic current $I_{ion}$, flowing across the membrane of a single cell, was mathematically described using the electrophysiological model of an adult mouse ventricular cardiomyocyte, first introduced by *Bondarenko et al., 2004*, including the model improvements in *Petkova-Kirova et al., 2012*. The model contains of 40 dynamical variables solved by a fourth-order Runge-Kutta method with the temporal resolution of $10^{-4}$ ms. Solving these variables describes 15 different currents as per *Equation 3*.

$$I_{ion} = I_{Na} + I_{CaL} + I_{pCa} + I_{Kto,f} + I_{Kto,s} + I_{Kr} + I_{Kur} + I_{Kss} + I_{K1} + I_{Ks} + I_{NaCa} + I_{NaK} + I_{Cl,Ca} + I_{Cab} + I_{Nab} \tag{3}$$

Here, $I_{Na}$ is the fast $Na^+$ current, $I_{CaL}$ is the L-type $Ca^{2+}$ current, $I_{pCa}$ is the $Ca^{2+}$ pump current, $I_{Kto,f}$ is the rapidly recovering transient outward $K^+$ current, $I_{Kto,s}$ is the slowly recovering transient outward $K^+$ current , $I_{Kr}$ is the rapid delayed rectifier $K^+$ current, $I_{Kur}$ is the ultrarapidly activating delayed rectifier $K^+$ current, $I_{Kss}$ is the non-inactivating steady-state voltage-activated $K^+$ current, $I_{K1}$ is the time-independent inwardly rectifying $K^+$ current, $I_{Ks}$ is the slow delayed rectifier $K^+$ current, $I_{NaCa}$ is the $Na^+/Ca^{2+}$ exchange current, $I_{NaK}$ is the $Na^+/K^+$ pump current, $I_{Cl,Ca}$ is the $Ca^{2+}$-activated $Cl^-$ current , $I_{Cab}$ is the background $Ca^{2+}$ current and $I_{Nab}$ is the background $Na^+$ current.

In spatially extended media, such as 2D, cardiac cells communicate with each other through intercellular coupling. The membrane voltage is then modeled using a reaction-diffusion type equation (*Equation 4*):

$$\frac{dV}{dt} = \nabla(D\nabla V) - \frac{I_{ion} + I_{stim}}{C_m} \tag{4}$$

The first term on the right side of the equation shows the intercellular coupling. $D$ is the diffusion tensor, which is assumed here to be a scalar and has the value 0.00014 cm/ms. In this 2D monodomain model, the excitation wave propagates with an isotropic conduction velocity of 43.9 cm/s. This simulation domain consists of 100 × 10 or 100 × 100 grid points. We use a spatial resolution of 0.025 cm and time step $10^{-4}$ ms. We apply no-flux boundary conditions at the inexcitable domain boundaries.

To create a spiral wave in the domain, we first selected four sets of values of the model parameters corresponding to four different phases of the AP: resting state, depolarized state and two states of repolarization (one at the beginning and one towards the end) (see *Figure 1—figure supplement 1A*). Next, we partitioned the domain into four sections and initialized all the dynamical variables with the values selected in the previous step (see *Figure 1—figure supplement 1B*). In clockwise direction, starting from the top left corner, we initialized the domain sections with values corresponding to the resting state, the depolarized state, beginning of the repolarization state and end

of the repolarization state, respectively. A plane wave then begins to propagate from the upper right to the upper left quarter of the domain. Since the lower half of the domain is refractory, the wave cannot propagate into it. However, with time, the lower left quarter recovers completely, allowing the wave to curl into it from above. Finally, the lower right quarter recovers for excitation and allows wave propagation into it, thereby completing the circuit for the spiral to travel through. *Figure 1—figure supplement 1D* shows a series of frames during the process of spiral wave formation.

To include light sensitivity, the model is coupled to the mathematical model of a light-activated protein called channelrhodopsin-2 (ChR2) (*Williams et al., 2013*). This protein is a non-selective cation channel that reacts to blue light with a wavelength of 470 nm. The inward ChR2 current ($I_{ChR2}$) is mathematically described by the following equation:

$$I_{ChR2} = g_{ChR2}G(V)(O_1 + \gamma O_2)(V - E_{ChR2}) \tag{5}$$

Here, $g_{ChR2}$ is the conductance, $G(V)$ is the voltage rectification function, $O_1$ and $O_2$ are the open state probabilities of the ChR2, $\gamma$ is the ratio $O_1/O_2$, and $E_{ChR2}$ is the reversal potential of this channel. By including the mathematical model of the ChR2 to this model, we can stimulate the system optically at the single cell level or the 2D monodomain level. In our studies, a stationary spiral had a circular core with constant curvature. Over time, the induction of drift led to a change in the curvature of the tip trajectory. The instantaneous curvature ($k$) of the spiral tip trajectory was calculated according to *Equation 6*, where ($x$, $y$) represents the coordinate of each point of the trajectory. For studies on spiral wave drift in the presence of gradient and stepwise illumination, we calculated maximum displacement $d$ at the end of 2 s, as mean of 10 different initial conditions of the spiral.

$$k = \frac{x'y'' - y'x''}{(x'^2 + y'^2)^{\frac{3}{2}}} \tag{6}$$

## Experimental study

All experiments in the intact mouse heart were done in accordance with the guidelines from Directive 2010/63/EU of the European Parliament on the protection of animals used for scientific purposes and the current version of the German animal welfare law and were reported to our animal welfare representatives. The experimental protocol was approved by the Italian Ministry of Health; authorization n° 944/2018 P and the responsible animal welfare authority (Lower Saxony State Office for Consumer protection and Food Safety). Humane welfare-oriented procedures were carried out in accordance with the Guide for the Care and Use of Laboratory Animals and done after recommendations of the Federation of Laboratory Animal Science Associations (FELASA). Key resources used for the bench research involving the intact mouse heart are provided in *Tables 2* and *3*.

## Experimental measurements of arrhythmia frequency in the intact mouse heart

To observe the effects of sub-threshold illumination on the arrhythmia frequency, we applied light globally to the hearts of Langendorff-perfused adult α-MHC-ChR2 transgenic mice (*Quiñonez Uribe et al., 2018*). The expression of channelrhodopsin-2 (ChR2) in these hearts was restricted to cardiomyocytes. For perfusion, we used the standard protocol of retrograde Langendorff perfusion with tyrode solution (130 mM NaCl, 4 mM KCl, 1 mM MgCl₂, 24 mM NaHCO₃, 1.8 mM CaCl₂, 1.2 mM KH₂PO₄, 5.6 mM glucose, 1% albumin/BSA, aerated with carbogen [95% oxygen and 5% CO2]). All experiments were performed at 37°C. Arrhythmia was induced by applying 30 electrical pulses (2.3–

**Table 2.** Key resource table for bench research involving the intact mouse heart (for studying the arrhythmia frequency).

| Reagent type | Designation | Source | Additional information |
|---|---|---|---|
| Biological sample | Transgenic mouse heart expressing ChR2 | Dr. S. Sonntag, PolyGene AG, Switzerland | Isonated form transgenic mouse (α-MHC-ChR2) |
| Chemical compound, drug | Di-4-ANBDQPQ stain | AAT Bioquest | Red-shifted voltage-sensitive dye to optically probe membrane potentials |
| Software, algorithm | AcqKnowledge | BIOPAC Systems, Inc | Software for Data Acquisition and Analysis |

**Table 3.** Key resource table for bench research involving the intact mouse heart (for studying the conduction velocity).

| Reagent type | Designation | Source | Additional information |
|---|---|---|---|
| Biological sample | Transgenic mouse heart expressing ChR2 | Prof. Marina Campione, University of Padova, Italy | Isolated from transgenic mouse (ChR2-MyHC6-Cre+) |
| Chemical compound, drug | Di-4-ANBDQPQ stain | Prof. Leslie M. Loew, Center for Cell Analysis and Modeling, UConn Health, Farmington (USA) | Red-shifted voltage-sensitive dye to optically probe membrane potentials |
| Software, algorithm | LabVIEW 2015 (64-bit) software HCImageLive software camera | National Instruments, Austin, TX, USA Hamamatsu, Shizuoka, Japan | — |

2.5 V amplitude), at frequencies of 30–50 Hz, using a needle electrode. To stabilize the arrhythmia, (*i*) the concentration of KCl in tyrode solution was reduced from 4 mM to 2 mM, and (*ii*), 100 µM Pinacidil, (a KATP channel activator) was added to the tyrode. To exclude the possibility of self-termination, we considered only those cases in which the arrhythmia lasted longer than 5 s. Next, a single blue light pulse ($\lambda$ = 470 nm, pulse duration = 1 s) was applied using 3 LEDs positioned at angular separation 120°, around the bath, to provide global illumination. We repeated the experiments for LI = 0.0011, 0.0041, 0.0078, 0.0124, and 0.0145 mW/mm$^2$, respectively, and measured the DF of the arrhythmia using a method of Fourier transform (FT) in those experiments that did not result in the termination of the arrhythmia. We considered data from two hearts with seven experiments on each.

## Experimental measurements of conduction velocity in the intact mouse heart

A wide-field mesoscope operating at a frame rate of 2 kHz (*Scardigli et al., 2018*) was used to map the action potential propagation in Langendorff horizontally perfused adult mouse hearts expressing ChR2 (under the control of $\alpha$-MyHC-ChR2 promoter) and stained with a red-shifted voltage sensitive dye (di-4-ANBDQPQ; *Matiukas et al., 2007*). To observe the effect of sub-threshold ChR2 stimulation on action potential conduction velocity, the heart was uniformly illuminated with blue light during electrical pacing at the heart apex (5 Hz). Conduction velocity was calculated by measuring the AP propagation time between two regions place at a known distance. The experiments were repeated for four different hearts at the LI = 0, 0.0104, 0.0175, 0.0262, 0.0894, 0.1528 mW/mm$^2$. We considered data from four hearts with three experiments on each.

## Acknowledgements

We thank Dr. Florian Spreckelsen, Babak Vajdi Hokmabad, and all the members of Research Group Biomedical Physics of Max Planck Institute for Dynamics and Self-Organization (MPIDS) for their fruitful input.

## Additional information

### Funding

| Funder | Grant reference number | Author |
|---|---|---|
| German Centre for Cardiovascular Research | | Stefan Luther |
| German Research Foundation | SFB 1002 Modulatory Units in Heart Failure | Stefan Luther |
| Max Planck Society | | Stefan Luther |

The funders had no role in study design, data collection and interpretation, or the decision to submit the work for publication.

## Author contributions
Sayedeh Hussaini, Conceptualization, Data curation, Software, Formal analysis, Validation, Investigation, Visualization, Methodology, Writing - original draft, Writing - review and editing; Vishalini Venkatesan, Data curation, Investigation, Methodology, Writing - original draft, Writing - review and editing; Valentina Biasci, José M Romero Sepúlveda, Data curation, Validation, Investigation, Visualization, Writing - review and editing; Raul A Quiñonez Uribe, Data curation, Investigation, Methodology; Leonardo Sacconi, Gil Bub, Resources, Software, Supervision, Validation, Investigation, Visualization, Project administration, Writing - review and editing; Claudia Richter, Resources, Investigation, Project administration, Writing - review and editing; Valentin Krinski, Conceptualization, Supervision, Validation, Investigation, Methodology, Writing - review and editing; Ulrich Parlitz, Conceptualization, Supervision, Investigation, Methodology, Writing - review and editing; Rupamanjari Majumder, Conceptualization, Supervision, Validation, Investigation, Writing - original draft, Writing - review and editing; Stefan Luther, Conceptualization, Resources, Software, Supervision, Funding acquisition, Validation, Investigation, Methodology, Writing - original draft, Project administration, Writing - review and editing

## Author ORCIDs
Leonardo Sacconi https://orcid.org/0000-0002-9320-5085
Gil Bub http://orcid.org/0000-0002-5304-0036
Ulrich Parlitz http://orcid.org/0000-0003-3058-1435
Rupamanjari Majumder https://orcid.org/0000-0002-3851-9225
Stefan Luther https://orcid.org/0000-0001-7214-8125

## Ethics
Animal experimentation: All experiments in the intact mouse heart were done in accordance with the guidelines from Directive 2010/63/EU of the European Parliament on the protection of animals used for scientific purposes and the current version of the German animal welfare law and were reported to our animal welfare representatives. The experimental protocol was approved by the Italian Ministry of Health; authorization n° 944/2018-P and the responsible animal welfare authority (Lower Saxony State Office for Consumer protection and Food Safety). Humane welfare- oriented procedures were carried out in accordance with the Guide for the Care and Use of Laboratory Animals and done after recommendations of the Federation of Laboratory Animal Science Associations (FELASA).

## Decision letter and Author response
Decision letter https://doi.org/10.7554/eLife.59954.sa1
Author response https://doi.org/10.7554/eLife.59954.sa2

# Additional files

## Supplementary files
• Transparent reporting form

## Data availability
All data generated or analysed during this study are included in the manuscript and supporting files.

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
