## [Decision Letter]

**Acceptance summary:**

At present there is limited understanding on how optogenetic depolarization of cardiomyocyte membrane potential can terminate cardiac arrhythmias, especially when using sub-threshold light intensities. The current paper presents a 2D computational model describing spiral wave dynamics in cardiac tissue, showing that optically induced spiral wave drift can lead to drift-induced collision with the boundary in favor of termination. It thus provides mechanistic insight into optogenetic defibrillation suggesting novel experimental strategies for terminating arrhythmias by illumination.

**Decision letter after peer review:**

Thank you for submitting your article "Drift and termination of spiral waves in optogenetically-modified cardiac tissue at sub-threshold illumination" for consideration by *eLife*. Your article has been reviewed by two peer reviewers, including Franziska Schneider-Warme as the Reviewing Editor and Reviewer #1, and the evaluation has been overseen by Didier Stainier as the Senior Editor.

The reviewers have discussed the reviews with one another and the Reviewing Editor has drafted this decision to help you prepare a revised submission.

Summary:

In the presented manuscript, Hussaini and coworkers study the effects of sub-threshold optogenetic depolarization on the dynamics of spiral (voltage) waves in murine ventricular tissue. To this end, they use a 2D computational model that enables them to test how different light intensities and spatial illumination patterns influence the spatio-temporal dynamics of the respective spiral wave, and whether they enable driving the core towards the boundary for spiral wave termination. The authors analyze three strategies for modulation of spiral wave drift: (1) light gradient across the entire simulation domain; (2) single-intensity illumination of half the domain; and, (3) a "multi-step adjusted pattern" protocol, where the stimulated region is gradually shrunk from half to one 20th of the domain.

While computationally predicted changes in CV and dominant frequency were qualitatively confirmed by "wet" experiments using ChR2-expressing cultured myocytes and Langendorff-perfused mouse hearts, all other experiments are pure in silico calculations so far lacking wet experimental testing.

The presented work is methodologically sound and explores an interesting new application of cardiac optogenetics, which may be of interest to readers of *eLife*. However, these exciting data are difficult to interpret due to major issues with clarity of language and presentation of data in the Results section. We have the following suggestions to improve the manuscript.

Essential revisions:

1) In the current state, the manuscript has well written passages with detailed explanations of the observed data, however, other passages are difficult to understand, either because of sloppy language, or because the data is insufficiently explained and put into context. Please revise the entire manuscript in order to improve presentation of results and clarity. Please find a detailed list of suggestions below:

a) Please explain how you define "efficiency" of spiral wave termination and "improvement of efficiency" e.g. in the sixth paragraph of the Results. What are the determining parameters and how did you optimize them? In line, what does "further improvement in optimization" in the Introduction refer to? Please be more precise.

b) Introduction: What does the "above mentioned concept of low-energy control" refer to? Could you please explain this in more detail?

c) "we use two-dimensional (2D) simulation domains containing optogenetically-modified adult mouse ventricular cardiomyocytes". No, this is imprecise language that will confuse many readers. The simulation domains do not contain myocytes, but they do represent myocytes in a reasonably realistic way.

d) Figure 1G: It is difficult to see anything on this image, neither cells nor other contrasts. Could you please replace the image or increase the contrast?

e) Results, third paragraph: Badly written paragraph. 5.86 mm^2 contradicts scale bar in image. "Beyond normal" – please be more precise.

f) The figures describing the main findings (Figures 2-4) are extremely information rich, but Figure 2B/C/E/F (and similar panels in Figures 3-4) are quite difficult to interpret. Which lines correspond to which time points (especially the lower purple/blue traces – are those from the first cycles of reentrant activity at the beginning of the simulation?) The reader has no means of assessing the specific times or understanding how they were chosen. The authors refer to these as quiescent periods, but this should be clarified. Is it one trace per rotation of the spiral wave? Please use a color scheme that can be easily distinguished by color-blind readers as well. Color scales like viridis [used in Figure 1—figure supplement 1] are generally more forgiving in this regard compared to Matlab jet [used in the figures discussed above].)

g) For all spatial representations of wave dynamics (e.g. Figure 2A/D) could you please include a scale bar?

h) "demonstrates a temporal drift" – please rephrase.

i) Please be more precise: e.g. "may or may not end up in stationarity" – what is the threshold? How can this be explained? "by increasing light intensity" – by which factor? (please also consider to revise in other places).

j) Please further explain Figure 4E/F. Please explain why (whether) the adjusted pattern is superior to the gradient pattern. Where do the orange lines come from and what exactly was supposed to be communicated here.

k) Discussion, last sentence of fourth paragraph: Incomplete sentence. Please also correct typos e.g. "optogenetics", "Langendorff".

l) Please consistently include space between numbers and units (missing in some sections).

m) "restricted to cardiac tissue"; strictly speaking: cardiomyocytes.

2) Could you please explain the observed differences between wet experimental findings and simulations, and discuss possible reasons for qualitative discrepancies (Figure 1)? Moreover, while the computational work in Figures 2-4 is quite convincing, the impact of the study would be greatly enhanced by addition of data from monolayers or intact hearts confirming these findings. Have you performed any wet experimental work confirming these results? If so, could you please include this supporting data? If no such data is available at present, please discuss possible future experiments to assess your in silico results.

3) Why did you use a 2.5 cm x 0.25 cm tissue geometry (stripe) in Figure 1A-D, especially when comparing the data to data obtained in isolated mouse hearts?

4) It seems to be a general finding that changes in V and dV/dx decline over time? Can you explain this finding? What are the underlying mechanisms (e.g. ChR inactivation, compensatory ionic currents?)? Do these values reach equilibrium (stationary state) and if so, at what time point is it reached?

5) The authors discuss their findings from the 2D models in the context of the decrease in light intensity when applying light by epicardial illumination in 3D (Discussion). While this is an interesting hypothesis, the reviewers disagree that the light-induced voltage gradient is approximately linear in small hearts. The estimated thickness of the mouse LV free wall during diastole is ~1 mm (see Saito et al., 2017), which is presumably even thinner in the RV; the approximate space constant for exponential decay of blue light in cardiac tissue is ~0.6 mm. It is also debatable whether the stimuli (long-durations at 0.4 mW/mm^2) used for mouse preparations in the Bruegmann et al., 2016 study induced transmural gradients with both supra- and sub-threshold regions. In fact, results from human heart simulations in Figure 5D of the same paper show that the effect of light attenuation at a depth of ~1mm vs. the illuminated epicardium is relatively negligible, although there are differences from the mouse experimental prep. Please further discuss your hypothesis by taking into account the dimensions of the mouse ventricle and previously applied light intensities.

6) Videos 2-4 are excellent and helped tremendously in the interpretation of Figures 2-4. Kudos to the authors on this! For Video 1, please explain why the light turning off seems to cause the same artefact as the light turning on. Is this correctly interpreted? In the same video, would it be possible to colorize the data so that the wave front is easier to track?

7) The technique used to induce spiral waves (described in the Materials and methods and highlighted in Figure 1—figure supplement 1): Is this a novel development for this study or has it been described previously? In the explanation the authors state that the sections are initialized with "four different values of cell membrane voltage". It would make more sense if ALL model state variables (esp. gating variables for the sodium channel) were initialized to the values corresponding to that part of the action potential. Otherwise, the spiral wave would most likely not occur because the cells in region four would not be refractory. Please explain further and correct, if needed.

8) Light intensities in wet experiments: Please describe how they were measured. How did you ensure homogenous illumination?

9) "spatial non-uniformity in the refractory period of cells that constitute the domain." This is an interesting finding; could the refractory period (x) be plotted?

---

## [Author Response]

[…] The presented work is methodologically sound and explores an interesting new application of cardiac optogenetics, which may be of interest to readers of eLife. However, these exciting data are difficult to interpret due to major issues with clarity of language and presentation of data in the Results section. We have the following suggestions to improve the manuscript.

We have made several additions and changes to manuscript. Following the reviewer’s suggestion, we have also revised the text to improve the clarity of the presentation.

Essential revisions:1) In the current state, the manuscript has well written passages with detailed explanations of the observed data, however, other passages are difficult to understand, either because of sloppy language, or because the data is insufficiently explained and put into context. Please revise the entire manuscript in order to improve presentation of results and clarity. Please find a detailed list of suggestions below:a) Please explain how you define "efficiency" of spiral wave termination and "improvement of efficiency" e.g. in the sixth paragraph of the Results. What are the determining parameters and how did you optimize them? In line, what does "further improvement in optimization" in the Introduction refer to? Please be more precise.

Spiral wave termination "efficiency" loosely refers to the ability of a structured illumination pattern to cause termination of the spiral wave by collision with the boundary due to higher drift velocity.

We have now included the following sentence to the Results section:

"to increase the drift velocity with which the spiral wave can approach the boundary."

We rephrased the line of our original manuscript to "Further progress in the clinical implementation of these developing techniques requires a deeper understanding of the underlying spiral and scroll wave dynamics." (Introduction).

b) Introduction: What does the "above mentioned concept of low-energy control" refer to? Could you please explain this in more detail?

We were referring to the idea of developing low-energy control schemes for the treatment of arrhythmias as alternative approaches to cardioversion or electrical defibrillation. However, we understand that the text may appear confusing. Thus, we have now rephrased it as follows:

"One low-energy technique to control arrhythmias in the clinical setting is Anti-Tachycardia Pacing (ATP)."

c) "we use two-dimensional (2D) simulation domains containing optogenetically-modified adult mouse ventricular cardiomyocytes". No, this is imprecise language that will confuse many readers. The simulation domains do not contain myocytes, but they do represent myocytes in a reasonably realistic way.

We thank the reviewer for this useful comment. We have now rephrased the sentence as follows:

"To demonstrate the drift of spiral waves using sub-threshold illumination, we use a two-dimensional (2D) continuum model of the cardiac syncytium, containing ionically-realistic representations of optogenetically-modified adult mouse ventricular cardiomyocytes at each node of the simulation domain.”

d) Figure 1G: It is difficult to see anything on this image, neither cells nor other contrasts. Could you please replace the image or increase the contrast?

Unfortunately due to the current situation, we couldn’t perform additional experiments of better resolution to improve the quality of the image in Figure 1F and Supplementary video 1. Since this data is not particularly adding extra value to the supporting simulations, we would like to avoid any obscurity in terms of presentation of the results in a clear and understandable manner. Thus, all co-authors have agreed to remove this single experiment from the paper.

e) Results, third paragraph: Badly written paragraph. 5.86 mm^2 contradicts scale bar in image. "Beyond normal" – please be more precise.

As mentioned in our response to comment 1d, we have now removed the data for the cell culture experiment. Consequently, we have also removed the following text from the manuscript:

"Similar behavior is observed in cell culture experiment of neonatal mouse heart expressed by ChR2(H134R). […] Thus the experimental data supports our finding that period of the spiral increases beyond normal in the presence of sub-threshold illumination."

"Beyond normal" simply indicates that application of sub-threshold light to the rotating spiral wave, increases its rotation period. The increased value is well above the mean rotation period of the spiral, in the absence of light. To increase clarity of the text, we have now removed the extra emphasis "Beyond normal".

f) The figures describing the main findings (Figures 2-4) are extremely information rich, but Figure 2B/C/E/F (and similar panels in Figures 3-4) are quite difficult to interpret. Which lines correspond to which time points (especially the lower purple/blue traces – are those from the first cycles of reentrant activity at the beginning of the simulation?) The reader has no means of assessing the specific times or understanding how they were chosen. The authors refer to these as quiescent periods, but this should be clarified. Is it one trace per rotation of the spiral wave? Please use a color scheme that can be easily distinguished by color-blind readers as well. Color scales like viridis [used in Figure 1—figure supplement 1] are generally more forgiving in this regard compared to Matlab jet [used in the figures discussed above].)

We thank the reviewer for pointing out this very relevant issue. We have now changed the colormap of the figures to viridis, as suggested by the reviewer. We have also modified the text corresponding to each figure as follows:

"Intrinsic inhomogeneity of cardiac tissue can cause a spiral wave to drift. Such inhomogeneity can be induced using sub-threshold illumination. In order to investigate the possibility of the induction of.…"

"In all the observed cases, the stationary spiral wave drifts towards the region with high resting membrane potential, which corresponds to the region with LI. […] We have ensured that each trace in Figures 2B and C represents the quiescent voltage profile of the domain at times which would correlate with successive turns of a spiral wave, drifting within the domain, in response to the establishing light-induced inhomogeneity."

"Figure 3A shows the migration of the spiral wave from interface toward the illuminated region during 2 s of the illumination. […] It shows that the inhomogeneity of the domain at the interface causes a spiral wave to drift, but this drift is eventually inhibited as the spiral migrates away from the interface."

"Figure 4B demonstrates the spatiotemporal response of the membrane voltage to the applied light pattern in a quiescent domain (no spiral wave), along the x-axis, at y=0.75 cm on the 2D domain, perpendicular to the illumination pattern during each step, shown as a dashed-dot line in Figure 4A."

g) For all spatial representations of wave dynamics (e.g. Figure 2A/D) could you please include a scale bar?

We have applied this change to all figures.

h) "demonstrates a temporal drift" – please rephrase.

We have now rephrased the sentence "To summarize, our results present substantial proof that the spiral wave demonstrates a temporal drift in response to uniform, global, constant sub-threshold illumination." as follows:

"To summarize, our results provide substantial evidence the change in spiral wave frequency, the so-called temporal drift, in response to uniform global constant sub-threshold illumination."

i) Please be more precise: e.g. "may or may not end up in stationarity" – what is the threshold? How can this be explained? "by increasing light intensity" – by which factor? (please also consider to revise in other places).

What we observe is the following: when a step-wise light pattern is applied to the domain with the spiral wave tip somewhere close to the border between illuminated and non-illuminated zones, a gradient of depolarization establishes over time at the border between the two zones. We call this region, the interface. Typically, a spiral wave drifts with high velocity when slope of the gradient at the interface is large. However, as the spiral migrates away from the non-illuminated region to enter the illuminated region, its drift velocity decreases exponentially with time. Once the spiral leaves the interface completely, to settle inside the illuminated region, it becomes stationary. However, we observe that the time required by the spiral to reach its stationary state also depends on the intensity of the applied light. Thus, we find that in some cases (greater than 0.015 mW/mm^2^), the spiral wave settles to zero drift velocity (stationary state) within 2 s of observation time, whereas, in other cases (smaller than 0.015 mW/mm^2^), the drift velocity decreases to a small non-zero constant value, within the 2 s of observation time, considered. We have now added the following text to the Results section:

"Typically, the drift velocity of the spiral tip in the initial phase is proportional to slope of the gradient at the interface. […] Thus, for the cases of LI *>* 0.015 mW/mm^2^, the spiral

wave settles to zero drift velocity (stationary state) within 2 s of observation time, whereas, in others, the drift velocity decreases to a small non-zero constant value, within the 2 s of observation time, considered."

We have rephrased the revised manuscript: "We found that by increasing LI from 0.001 to 0.015 mW/mm^2^, the initial velocity *V*_0_ increases by factor of 20.…"

j) Please further explain Figure 4E/F. Please explain why (whether) the adjusted pattern is superior to the gradient pattern. Where do the orange lines come from and what exactly was supposed to be communicated here.

To compare the spiral wave termination efficiency from the various applied protocols, such as gradient illumination with light intensity (LI) ranging from 0.0 to 0.01 mW/mm^2^ (Figure 2A) and multi-step adjusted pattern illumination with constant LI = 0.01 mW/mm^2^ (Figure 4A), we calculated the mean squared displacement (MSD) of the spiral wave core during 2 s of illumination (Figure 4E). Our studies showed that the MSD for the case of gradient illumination was *≈* 0.25 cm^2^, which is far from the boundary, whereas, that in the case of multi-step adjusted pattern illumination, was 1.25 cm^2^. The red dashed line represents the maximum MSD from the boundary. The drift velocity of the spiral wave core in each case, is shown in Figure 4F. Our results indicate that this parameter is *'* 4x higher in case of illumination via multi-step adjusted pattern, as compared to that measured with gradient illumination. Each peak for the case of multi-step adjusted pattern illumination corresponds to the instant when the illumination pattern is adjusted to the next step. Such adjustment causes the spiral wave to drift continuously towards the boundary and terminate within 2 s of simulation. Thus, the multi-step adjusted pattern of illumination proves to be more efficient at effectuating drift-induced termination of spiral waves, than the gradient illumination pattern.

We have now included the following paragraph to the explanation for Figure 4:

"Finally, to compare the efficiency of spiral wave termination from different illumination protocols, such as illumination with a smooth gradient in light intensity (LI) from 0.0 to 0.01 mW/mm^2^ (Figure 2A) and multi-step adjusted pattern illumination with constant LI = 0.01 mW/mm^2^ (Figure 4A), we calculated the mean square displacement (MSD) of the spiral wave core during 2 s of illumination (Figure 4E). […] Thus, the multi-step adjusted pattern proves to be more efficient at effectuating drift-induced termination of spiral waves, than the gradient illumination pattern."

k) Discussion, last sentence of fourth paragraph: Incomplete sentence. Please also correct typos e.g. "optogenetics", "Langendorff".

We have attended to this change.

l) Please consistently include space between numbers and units (missing in some sections).

We have attended to this change.

m) "restricted to cardiac tissue"; strictly speaking: cardiomyocytes.

We agree. We have changed this.

2) Could you please explain the observed differences between wet experimental findings and simulations, and discuss possible reasons for qualitative discrepancies (Figure 1)? Moreover, while the computational work in Figures 2-4 is quite convincing, the impact of the study would be greatly enhanced by addition of data from monolayers or intact hearts confirming these findings. Have you performed any wet experimental work confirming these results? If so, could you please include this supporting data? If no such data is available at present, please discuss possible future experiments to assess your in silico results.

The wet lab experiments were performed at three different places (Montreal, Florence and Göttingen), under different environmental conditions. Thus, the electrophysiological measurements, such as action potential recordings, conduction velocity etc. showed tolerable differences from each other. However, the model used for all simulations, namely, the Bondarenko model for adult mouse heart, turned out to be not the best possible representation of the experimental system. For example, the typical duration of an action potential, as measured from cell culture experiments, was 30 ms, whereas in the intact heart, we measured an APD_90_ of 60 ms. The same parameter measured 16 ms in simulations. In addition, we did not take into consideration, the natural cellular heterogeneity or the geometry of the system, i.e., the curvature of the surface of the intact heart, the fiber structure etc. Thus, our computer model was inherently different from the models used in experiments. One way to address this problem would be to tune the Bondarenko model parameters so as to match the morphology of the action potential with experiments. However, because of the Covid-19 situation, we were unable to perform further experiments to validate our findings. Hence, we continued using the original Bondarenko model for our studies. This model is time-tested and has been validated on multiple occasions by different research groups for different sets of studies [1, 2].

One possible reason for the qualitative differences between simulation and wet experimental data in the insets of Figure 1B and C, could be the difference in the system itself; we performed simulations in 2D (isotropic domain, with no fiber structure, no natural cellular heterogeneity and open boundaries), whereas the experiments were performed on the intact heart (anisotropic domain with fiber orientation, natural cellular heterogeneity and different domain topology). Thus, our experimental observations were found to deviate quantitatively from numerics. The main reason for including the experimental data alongside the numerical data in these sub-figures (1B and 1C) was to show that the trends were comparable.

In line with the reviewer’s suggestion, we have now modified the last paragraph of the Discussion section as follows:

“We propose to test this hypothesis in cell culture experiments by projecting a gradient sub-threshold light pattern on a monolayer of optogenetically-modified mouse cardiac cells. […] Such a drift occurs in the direction of the positive gradient, resulting in the emergence of the spiral cores on the surface, where they are eliminated through synchronisation."

3) Why did you use a 2.5 cm x 0.25 cm tissue geometry (stripe) in Figure 1A-D, especially when comparing the data to data obtained in isolated mouse hearts?

The reviewer has a very valid question. Ideally, we should have used an anatomically realistic 3D mouse model (which includes the fiber structure and curved topology in addition to a larger size) to perform these simulations, if the goal was to use the computer model to make predictions that could be validated in experiments. Here, however, our only concern was to ensure that the model shows the same qualitative results in important electrophysiological properties of the system that determine the type of spiral wave dynamics that the system can support. In this case, we tested the light-induced response in conduction velocity and dominant frequency of the spiral wave.

4) It seems to be a general finding that changes in V and dV/dx decline over time? Can you explain this finding? What are the underlying mechanisms (e.g. ChR inactivation, compensatory ionic currents?)? Do these values reach equilibrium (stationary state) and if so, at what time point is it reached?

Our studies show that application of light to an optogenetically-modified cardiomyocyte induces a nonlinear response in the membrane voltage. As the reviewer correctly observes, both V and dV/dx decrease over time. However, the rate of decrease is not constant. It is large at the initial stage, allowing an instantaneous (switch-like) response to light application, but declines gradually as a steady state is approached. This is shown in the context of our system in the inset of Figure 3C. We believe that this behaviour of the voltage response can be attributed to the kinetics of the Channelrhodopsin (ChR2) current. Application of light to an excitable cardiac cell membrane causes an initial spike in the ChR2 current. This spike results in a sharp depolarization of cell membrane, which very soon repolarises as the ChR2 current decays, shown in Author response image 1. In our system, at the timescales that we consider, we observe establishment of the stationary state in the gradient illumination case (unless the spiral drifted out of the domain already). In the case of single step illumination, the V value reached a stationary state within 2s (our observation time), but dV/dx continued to decrease further. For the case of multi-step illumination, however, we did not wait for the equilibrium to be reached in any step. Rather, we used the initial kick provided by the ChR2 current to shift the spiral closer and closer to the boundary for termination.

**Author response image 1. sa2fig1:** (**A**) Kinetics of the ChR2 current at different LI. All cases shows an initial peak which decays to the steady-state and when the light turns off decays to the baseline. (**B**) Time evolution of the voltage distribution along the illumination pattern with an exponential decay in a quiescent domain with the LI of 0.07 mW/mm^2^. (**C**) Spatial derivative of *V* (*dV/dx*) along the illumination pattern with an exponential decay at different times, for the same illumination as in (A).

5) The authors discuss their findings from the 2D models in the context of the decrease in light intensity when applying light by epicardial illumination in 3D (Discussion). While this is an interesting hypothesis, the reviewers disagree that the light-induced voltage gradient is approximately linear in small hearts. The estimated thickness of the mouse LV free wall during diastole is ~1 mm (see Saito et al., 2017), which is presumably even thinner in the RV; the approximate space constant for exponential decay of blue light in cardiac tissue is ~0.6 mm. It is also debatable whether the stimuli (long-durations at 0.4 mW/mm^2) used for mouse preparations in the Bruegmann et al., 2016 study induced transmural gradients with both supra- and sub-threshold regions. In fact, results from human heart simulations in Figure 5D of the same paper show that the effect of light attenuation at a depth of ~1mm vs. the illuminated epicardium is relatively negligible, although there are differences from the mouse experimental prep. Please further discuss your hypothesis by taking into account the dimensions of the mouse ventricle and previously applied light intensities.

We thank the reviewer for this comment. Based on the information in the introduced papers by the reviewers, we computes further simulations to show that the exponential decay of the blue light through the tissue induces the drift of the spiral wave. To do this we applied a global gradient illumination pattern where there is an exponential decay with a attenuation coefficient of 0.6 mm. Video 4 shows the drift of the spiral wave where the wave is at the neighbouring of the illumination region. Figure 2—figure supplement 1A and B show the voltage and voltage gradient changes across a line in a quiescent domain with a realistic exponential decay pattern of the illumination. We have added this simulation study to support our hypothesis mentioned in the Discussion section. We have now changed the discussion of this hypothesis as follows:

"Thus, our study provides new mechanistic insights into the theory of successful optogenetic defibrillation in animal hearts, where application of light to the surface produces a gradient of illumination within the heart wall. […] Video 5 shows a spiral wave rotation in a circular trajectory where there no illumination pattern".

6) Videos 2-4 are excellent and helped tremendously in the interpretation of Figures 2-4. Kudos to the authors on this! For Video 1, please explain why the light turning off seems to cause the same artefact as the light turning on. Is this correctly interpreted? In the same video, would it be possible to colorize the data so that the wave front is easier to track?

In optogenetics experiments, one of the most difficult tasks is to analyse optical mapping data that is measured in the presence of stimulating light. For these analyses, the type of spatial and temporal filtering that is applied to the optical mapping signal is typically different from that applied to the case without light. The ’bleaching’ effect of the light at the frames occurring at the initial and final stages of the light application is a common artefact of filtering (with high signal-to-noise ratio). thus we see the same effect when light is turned on or off.

However, as discussed above in comments 1d and 1e, we have now removed the cell culture data from the manuscript because of its lack of clarity and our inability to provide better quality images/videos, given the current situation. Thus, we have removed this video.

7) The technique used to induce spiral waves (described in the Materials and methods and highlighted in Figure 1—figure supplement 1): Is this a novel development for this study or has it been described previously? In the explanation the authors state that the sections are initialized with "four different values of cell membrane voltage". It would make more sense if ALL model state variables (esp. gating variables for the sodium channel) were initialized to the values corresponding to that part of the action potential. Otherwise, the spiral wave would most likely not occur because the cells in region four would not be refractory. Please explain further and correct, if needed.

We thank the reviewer for this insightful comment. First of all this is a simple numerical method used to induce spiral waves in generic, as well as complex ionic models As the reviewer has correctly pointed out, all the dynamical variables of this model should be initialized by the values corresponding to the chosen parts of the action potential. We have now changed the corresponding text in the Materials and methods section as follows:

“To create a spiral wave in the domain, we first selected four sets of values of the model parameters corresponding to four different phases of the AP: resting state, depolarized state and two states of repolarization (one at the beginning and one towards the end) (see Figure 1—figure supplement 1A). […] Figure 1—figure supplement 1D shows a series of frames during the process of spiral wave formation."

8) Light intensities in wet experiments: Please describe how they were measured. How did you ensure homogenous illumination?

In order to calibrate and ensure homogeneous illumination in the ex-vivo experiments of Langendorff perfused hearts, to study the effect of low LI on the arrhythmia frequency, the following was procedure was performed prior to each experiment using the PM100D optical power meter and the S120VC photodiode power sensor (Thorlabs): The power sensor would be placed at the position and distance the heart would occupy during the experiment. Different intensities would be measured facing each of the three different LEDs surrounding the heart. This would be repeated in order to ensure that the power sensor registered the same amount of light from the three LEDs and achieve homogeneity during light stimulation. Since the heart does not have a homogeneous shape, we can expect minor differences that in general can represent a challenge in any optogenetic experiment at an organ level.

For the experimental study of the CV in intact mouse heart, the absolute Laser intensity was measured after the objective lens by using a standard photodiode sensor (PD300-3W-V1,Ophir). The illumination homogeneity across the whole heart was measured by using an sCMOS camera (ORCA-flash 4.0, Hamamatsu). We found a coefficient of variation(STD/AVERAGE) of the intensity of the order of

0.2.

For the cell culture experiment the light intensity was calibrated to ensure homogeneous illumination using a PM100D power meter and S130C photodiode power sensor measuring direct light beam intensity (mW/mm^2^). We are using around 4% of the sensor area and we expect minimal variation of light intensity over such a small portion of the projector’s FOV.

9) "spatial non-uniformity in the refractory period of cells that constitute the domain." This is an interesting finding; could the refractory period (x) be plotted?

Thanks for the comment. To show this effect we applied a gradient LI of (0, 0.01) mW/mm^2^ in a 2D domain. Then we initiated a planar wave along the illumination gradient by applying an electrical stimulation to a region of the left-hand side of the domain. Author response image 2 shows a time-series of the voltage at 20 different points in the domain along the illumination gradient. To show precisely the effect of the LI on the refractory period we looked at the APD_75%_, shown in Author response image 2. This showing an increase of the APD confirming the non-uniformity of the refractory period along the gradient.

**Author response image 2. sa2fig2:** (**A**) It indicates the action potentials at 15 different points at the domain with a linear gradient pattern when a planar wave propagates through the domain perpendicular to the illumination pattern, shown in (**B**). The inset shows an increase of APD_75_ along the illumination gradient showing the non-uniformity of the refractory period along the gradient.

**References:**

1) L Li, S A Niederer, W Idigo, Y H Zhang, P Swietach, B Casadei, N P Smith A mathematical model of the murine ventricular myocyte: a data-driven biophysically based approach applied to mice overexpressing the canine NCX isoform, Am J Physiol Heart Circ Physiol. 2010 Oct;299(4):H1045-63.

2) Jason H. Yang and Jeffrey J. Saucerman, Phospholemman is a Negative Feed-Forward Regulator of Ca^2+^ in beta-Adrenergic Signaling, Accelerating beta-Adrenergic Inotropy, J Mol Cell Cardiol. 2012 May;52(5):1048-55.